# Melatonin Moderates the Triangle of Chronic Pain, Sleep Architecture and Immunometabolic Traffic

**DOI:** 10.3390/biomedicines9080984

**Published:** 2021-08-09

**Authors:** Shafqat R. Chaudhry, Andreas Stadlbauer, Michael Buchfelder, Thomas M. Kinfe

**Affiliations:** 1Department of Basic Medical Sciences, Shifa College of Pharmaceutical Sciences, Shifa Tameer-e-Millat University, Islamabad 44000, Pakistan; Shafqat.Rasul@yahoo.de; 2Department of Neurosurgery, Medical Faculty, Friedrich-Alexander University (FAU) of Erlangen-Nürnberg, D-91054 Erlangen, Germany; Andreas.Stadlbauer@uk-erlangen.de (A.S.); Michael.Buchfelder@uk-erlangen.de (M.B.); 3Division of Functional Neurosurgery and Stereotaxy, Medical Faculty, Friedrich-Alexander University (FAU) of Erlangen-Nürnberg, D-91054 Erlangen, Germany

**Keywords:** melatonin, chronic pain, sleep, immune-metabolism, central and peripheral neuroinflammation pathways, augmentation of neurostimulation therapies

## Abstract

Preclinical as well as human studies indicate that melatonin is essential for a physiological sleep state, promotes analgesia and is involved in immunometabolic signaling by regulating neuroinflammatory pathways. Experimental and clinical neuromodulation studies for chronic pain treatment suggest that neurostimulation therapies such as spinal cord stimulation, vagus nerve stimulation and dorsal root ganglion stimulation have an impact on circulating inflammatory mediators in blood, cerebrospinal fluid and saliva. Herein, we provide an overview of current literature relevant for the shared pathways of sleep, pain and immunometabolism and elaborate the impact of melatonin on the crossroad of sleep, chronic pain and immunometabolism. Furthermore, we discuss the potential of melatonin as an adjunct to neurostimulation therapies. In this narrative review, we addressed these questions using the following search terms: melatonin, sleep, immunometabolism, obesity, chronic pain, neuromodulation, neurostimulation, neuroinflammation, molecular inflammatory phenotyping. So far, the majority of the published literature is derived from experimental studies and studies specifically assessing these relationships in context to neurostimulation are sparse. Thus, the adjunct potential of melatonin in clinical neurostimulation has not been evaluated under the umbrella of randomized-controlled trials and deserves increased attention as melatonin interacts and shares pathways relevant for noninvasive and invasive neurostimulation therapies.

## 1. Introduction

Melatonin, an indole alkaloid, is chemically N-acetyl-5-methoxytryptamine and is a derivative of tryptophan [1]. Melatonin, acting as a pleiotropic hormone, is released from the pineal gland and extra-pineal tissues with a critical role in regulating the circadian rhythms [2]. Initially, melatonin was described as a sleep hormone as it is secreted in the dark and induces sleep, but it is now well appreciated that it implicates a wide array of activities encompassing anti-oxidant, anti-inflammatory, anti-apoptotic, anti-sympathetic nerve activation, endothelial cell preservation, neuroprotection, hepatoprotection, immunomodulation, thermoregulation, mood and sexual behavior modulation [1,3,4,5,6,7,8]. Although the precise mechanism remains to be elaborated, these implicates a relevant role of the neurobiology of a broad variety of system inflammation driven diseases such as chronic pain syndromes and metabolic associated disorders.

Hence, melatonin has the capability to promote and to interact with relevant and shared pathways of sleep, chronic pain and immunometabolic trafficking due to its anti-inflammatory properties. Melatonin derives its name due to its ability to induce aggregation of melanin granules, which lead to lightening of the skin of the frog due to its α-Melanocyte Stimulating Hormone (α-MSH) antagonistic effects [4,9]. Besides pineal body, different brain areas, retina, retinal pigment epithelium, lens, iris, ciliary body, lacrimal glands, bone marrow, lymphocytes, skin and gut secrete it [4,9,10] and is conserved across almost all species from primitive prokaryotes to humans [4]. Serum levels of melatonin tend to rise after a few months of birth in neonates, gradually decrease during puberty and eventually decline at old age characterized by rhythmic diurnal release variations exhibiting peak levels at night from midnight to 2 am and lowest from midday to 2 pm [4,10].

Over the recent years, non-invasive and invasive neurostimulation of both the central and peripheral nervous systems have become established therapy options in drug-resistant chronic pain disorders of different origins including but not limited to spinal cord stimulation, dorsal root ganglion stimulation and/or vagus nerve stimulation [11,12,13,14,15,16,17,18,19,20,21,22,23]. So far, outcome of neuromodulation therapies for chronic pain has been determined using score-based parameters and psychometric features (depression, cognition decline, anxiety) neglecting the intra- and inter-individual variability of chronic patients. In view of lacking objective outcome parameters, several uncontrolled and randomized clinical studies have determined the effects of neurostimulation therapies on the neuro-immune axis by quantifying the concentrations of circulating inflammatory mediators and corresponding gene expression levels in different biofluids (blood, cerebrospinal fluid, saliva) in chronic pain patients [24,25,26,27]. Intriguingly, several approaches such as molecular inflammatory phenotyping, electrophysiological recordings, neuroimaging and digital measures have gained increased recognition in order to predict and potentially consolidate the neuromodulation therapy outcome. However, systematic evidence-derived studies assessing the augmentative potential of adjunct anti-inflammatory pharmacological agents such as melatonin, which can modulate the inflammatory pathways, remain under-investigated.

Herein, we provide an overview addressed to melatonin neurobiology, its anti-inflammatory capabilities, and its complex and dynamic interplay with sleep physiology, chronic pain genesis/maintenance and deteriorated metabolism, which in turn represent common clinical characteristics of chronic pain individuals. Furthermore, we discuss the possibility and limitations of melatonin as an adjunct therapy to enhance neurostimulation responsiveness for chronic pain disorders

## 2. Materials and Methods

A literature review was performed in several databases such as PubMed, MEDLINE/OVID, and SCOPUS along with manual searches of original and review articles from 1990–2020. The following terms: melatonin, chronic pain, sleep, immune-metabolism, neuroinflammation, biomarkers, metabolic disorders and augmentation of neurostimulation therapies were used to identify relevant preclinical and in–human derived literature. Chronic pain studies assessing and combining neurostimulation with melatonin are lacking, while there is exist a large number of publications addressed to melatonin neurobiology indicating melatonin to play a crucial role for the pathophysiology of chronic pain, sleep and metabolism. All these characteristics are disrupted frequently in chronic pain patients. In particular, the impact and adjunct value of melatonin for neurostimulation therapies intended to treat chronic pain disorders remains largely unknown.

Due to the limited data available pertaining to the particular pain types along with outcome parameters, the findings were presented in the form of a narrative review.

## 3. Overview of the Melatonin biology

### 3.1. Synthesis and Metabolism of Melatonin

Tryptophan serves as a precursor of Melatonin. Initially, tryptophan is converted into serotonin, and then serotonin is *N*-acetylated and *O*-methylated to form melatonin by arylalkylamine *N*-acetyltransferase (NAT) and hydroxyindole-*O*-methyltransferase (HIOMT), respectively [28]. The suprachiasmatic nucleus of the hypothalamus regulates the synthesis of melatonin by the pineal gland depending upon the input received from retinohypothalamic pathway. During day, the exposure to light induces the breakdown of melanopsin in retinal photoreceptive ganglion cells, which signals through retinohypothalamic pathway to inhibit the synthesis of melatonin, consequently, the levels of melatonin are low during daylight and rise in dark at night [3]. The neurotransmitters such as noradrenaline and GABA regulate the synthesis of Melatonin [29]. The presence of 5-methoxy and N-acetyl functional groups in the structure of melatonin impart specificity and amphiphilicity, and melatonin can readily cross the plasma membranes affecting almost all the cells of the body [3]. In the blood after release from the pineal body, it is transported in combination with albumin and is metabolized in the liver through oxidation by Cytochrome P450 isozymes (CYP1A2, CYP1A1, CYP1B1) into 6-hydroxymelatonin and subsequent conjugation to 6-sulfatoxymelatonin, that is ultimately eliminated via kidneys [3,28,30]. At the level of brain, substantial quantity of melatonin is biotransformed into kynuramine derivatives, which share antioxidant and anti-inflammatory properties with melatonin (Figure 1) [30].

### 3.2. Melatonin and Its Receptors

The extensive accessibility throughout the body and versatility in the effects of melatonin is brought about by its receptor dependent and independent effects. The anti-oxidative and free radical scavenger effects of melatonin avoids any receptor based interaction and directly stem from melatonin’s direct chemical interaction with various free radicals such as hydroxyl radicals, hydrogen peroxide, nitric oxide and others [9]. Melatonin is known to upregulate the expression or activity of different antioxidant enzymes such as superoxide dismutase and glutathione peroxidase [3]. In addition to these directly mediated effects, two high affinity membrane bound receptors are recognized to mediate the effects of melatonin and are designated as MT1 and MT2. Both MT1 and MT2 belong to the family of G-protein coupled transmembrane serpentine receptors (GPCRs) [3,9,31]. A relatively low binding affinity site MT3 for melatonin was in addition recognized intracellularly on a cytosolic enzyme quinone reductase 2, which is responsible for reducing menadione and other quinones [3,31]. Furthermore, melatonin binds nuclear receptors such as retinoid orphan receptors (ROR) and retinoid Z receptors (RZR) such as RORα, RORα2, RZRα and RZRβ [3,9].

#### MT1 and MT2 Receptors

The MT1 receptor, cloned in 1994 as a first mammalian melatonin receptor, consists of 350 amino acids and is linked to G_i_, more specifically G_iα2_, G_iα3_ and G_q/11_ [9,32]. The activation of the MT1 receptor leads to inhibition of adenylyl cyclase leading to a decline in cAMP levels. The MT1 receptors can be coupled to Kir3 potassium channels through G_i_ and in some cells to G_q/11_, mediating the activation of phospholipase C resulting in increased Ca^2+^ levels through modulation of the calmodulin levels [9,31].

The MT1 receptor is found to be expressed in brain, peripheral blood vessels, aorta, heart, testes, ovaries, placenta, mammary glands, skin, liver, spleen, kidney, adrenal cortex, pancreas, and immune system. Inside the brain, expression of MT1 receptors is specifically found in hypothalamus, hippocampus, cerebellum, pars tuberalis, substantia nigra and ventral tegmental area (Figure 1) [3,9,31].

The MT2 receptor shares 60% homology to MT1 receptor and is composed of 363 amino acids. The MT2 receptor like MT1 receptor is coupled to G_i_ and leads to inhibition of adenylyl cyclase and soluble guanylyl cyclase activity [9,31]. The MT2 receptors aredistributed in the gastrointestinal tract, skin, breasts, testes, kidney, adipose tissue, immune system, retina, blood vessels, and inside the brain including the regions such as hypothalamus, suprachiasmatic nucleus and pituitary [9]. Both the membrane receptors have been shown to dimerize forming homodimers or heterodimers, with relatively more common homodimers of MT1 or MT1-MT2 heterodimers compared to MT2 homodimers [9]. Interestingly, both the receptors may have opposing physiological functions e.g., non-rapid eye movement (NREM) sleep is decreased upon MT1 activation and promoted upon MT2 activation; MT1 activation leads to vasoconstriction, but MT2 activation produce vasodilation; MT1 receptor have implications in depression and anhedonia, whereas MT2 receptor is implicated in anxiety; further MT1 receptors regulate rapid-eye movement (REM) sleep, but MT2 receptors do not regulate it [32].

## 4. Melatonin and Chronic Pain

Pain has been defined as “An unpleasant sensory and emotional experience associated with actual or potential tissue damage or described in terms of such damage” [33]. Pain significantly affects the quality of life and implies increased social costs for its management [34]. A relationship between pain and melatonin has beendelineated, since chronic pain patients have shown lower levels of melatonin in the blood and urine [35]. Even the precursors of melatonin such as L-tryptophan and serotonin were found to be low in patients with fibromyalgia, indicating that the melatonin has a critical role in chronic pain syndromes [35,36]. Further adding to the evidence, analgesia observed after melatonin administration in both nociceptive and neuropathic pain models also points towards an important role of melatonin in regulating pain [37]. Melatonin affords amelioration of pain through several mechanisms. For instance, Melatonin could ameliorate the nociceptive pain owing to its ability to prevent prostaglandins release, inhibition of the migration of polymorphonuclear cells at the inflammation site, and inhibition of the cyclooxygenase-2 (COX-2) and nitric oxide (NO) synthase. Melatonin provides neuropathic pain relief through its ability to raise the pain threshold and decrease of the thermal hyperalgesia manifestation. Intriguingly, the analgesic effects of melatonin have been shown to be dose dependent and have been observed consistently across various pain induction models relying on thermal, chemical, mechanical and electrical pain induction [37]. Furthermore, the different routes of administration almost provide the same efficacy from a given dose of the melatonin [37]. Analgesic effects of melatonin have been well appreciated across various chronic pain syndromes such as chronic pelvic pain, fibromyalgia, irritable bowel syndrome, tension and cluster headaches, migraine and others including chronic back pain [37,38]. Around 5–10% of the patients with acute back pain progress to subacute back pain category and then, finally develop chronic back pain, which is defined as back pain lasting for more than 3 months [34]. The next section will provide an overview of the melatonin’s pain-relieving mechanisms before discussing the melatonin’s effects in chronic pain disorders of different origin.

### 4.1. Mechanisms of Melatonin Action in Relieving Pain

The pain-relieving effects of melatonin either stem from its direct anti-nociceptive effects due to its anti-oxidant, anti-inflammatory properties and direct interaction with its cognate receptors or through indirect complex interactions with other systems that can modulate the pain perception pathways [29,37].

MT2 receptors have been found to be particularly enriched in the brain areas which play an important role in modulating the pain sensation [29]. For example, ventromedial nucleus of the thalamus and reticular formation as a part of the ascending nociceptive pathway and ventrolateral periaqueductal gray matter as a part of the descending antinociceptive pathway harbor high densities of the MT2 receptors [32]. Initial studies involving the use of melatonin antagonists also confirmed that MT2 receptors are preferentially implicated in mediating melatonin’s anti-nociceptive effects [32]. Furthermore, selective partial agonists at MT2 receptors such as UCM765 and UCM924 have shown anti-nociceptive effects in experimental models of nociceptive pain [29,37]. Modulation of the MT2 receptors expressed on glutamatergic neurons in the rostral ventrolateral periaqueductal gray by UCM924 decreased tail flick responses in experimental pain model and modulated the neuronal activity of the pain pathways [29,37]. An agonist of MT2 receptor, IIK-7, provides relief against neuropathic pain via inhibition of glial cells activation and inhibition of inflammation as exhibited by down regulation of inducible nitric oxide synthase (iNOS) and caspase 3 [29]. Intriguingly, MT2 receptor ligation by melatonin reduces neuronal excitability in experimental pain model owing to inhibition of T-type Ca^2+^ channel currents via downstream G_βγ_-mediated PKCη signal pathway (Figure 1). In addition, melatonin suppresses mechanical allodynia and thermal hyperalgesia due to the inhibition of the mitogen activated protein kinase (MAPK) and calcium signaling [29].

### 4.2. Interaction with Neurotransmitters (and Their Receptors)

Melatonin interacts with various neurotransmitter systems indirectly to produce nociceptive effects. For instance, melatonin can produce antinociceptive effects by affecting GABAergic (GABA_A_ receptor), benzodiazepinergic (BZD receptor), opioidergic (δ/µ/κ receptors), sigma system (sigma receptor), dopaminergic receptor (D_2_ receptor), adrenergic (α_2_ receptor), serotonergic (5-HT_2A_ receptor), glutamatergic (NMDA receptor) and NO-cyclic GMP-PKG signaling pathway (Figure 1) [30].

Melatonin can modulate the GABAergic transmission as was evidenced from the observation that Flumazenil, which is a benzodiazepine antagonist and inhibits indirectly the GABAergic neurotransmission via antagonizing the allosteric effects of benzodiazepines receptors on their own receptor subtypes and therefore reversing the anti-nociceptive effects of Melatonin [32,37,38,39]. Melatonin is known to facilitate GABAergic transmission through various mechanisms such as the increase of the GABA content, increased affinity of GABA for its receptors and density of GABA receptors; even melatonin and its analogues can directly bind to GABA receptors [31,32,37,38,39]. These mechanisms likely explain the synergistic effects of melatonin and benzodiazepine-GABA agonists, which are independent from direct interaction of melatonin to its cognate receptors (Figure 1).

Interestingly, there is evidence that there exists an indirect link between melatonin and the opioidergic system. For instance, the melatonin’s antinociceptive effects were antagonized upon Naloxone administration, which is an opiate antagonist [37,38]. Further, melatonin administration could induce the production and release of β-endorphins; and morphine and other opioid agonists could induce the melatonin release from the pineal gland [39]. Furthermore, the analgesic effects of morphine were reduced in pinealectomized mice. Intriguingly, neither melatonin nor any of its analogues were shown to directly interact with opioid receptors, yet there exists an important connection between melatonin and opioids in mediating anti-nociceptive effects [37,38,39].

Glutamate and its NMDA receptors have been implicated in nociceptive sensation pathways [29]. Repetitive firing of C-fibers potentiates the spinal nociceptive transmission, an effect known as wind-up, and is dependent on NMDA receptors. Interestingly, melatonin can induce suppression of this wind-up effect in a dose dependent manner through the modulation of intracellular NO levels leading to the activation of cGMP-PKG-ATP sensitive K^+^ channels [30]. The antinociceptive effects of melatonin involve modulation of the α_1_-adrenergic, α_2_-adrenergic, muscarinic and nicotinic receptors at the spinal level through the modulation of intracellular cGMP levels (Figure 1) [30].

### 4.3. Modulation of Ion Channel Activity

Melatonin has been shown to activate the G-protein coupled Kir3 (inwardly rectifying K^+^) channels at the cellular level, which reduce the rapid firing of action potential trains in neurons [37,38]. Melatonin activates outward flow of K^+^ ions in various areas of the CNS such as cerebellum, suprachiasmatic nucleus , while in turn it inhibits the activity of voltage gated sodium channels such as Nav1.8 and Nav1.9 to exert anti-thermal hypersensitivity and anti-mechanical allodynia effects [29,37,38,39]. In the cultured dorsal root ganglion cells of the spinal cord, melatonin has been shown to decrease the neuronal free intracellular Ca^2+^ levels via inhibition of the voltage sensitive Ca^2+^ channels [37,38,39]. This effect of melatonin to inhibit mobilization of free Ca^2+^ from intracellular storage sites as well as inhibition of inward flow of Ca^2+^ from voltage activated channels has also been demonstrated in various other tissues [29]. Melatonin regulates against the Ca^2+^ influx through the desensitization of transient receptor potential vanilloid type 1 (TRPV1) and transient receptor potential melastatin type 2 (TRPM2) channels (Figure 1) [29].

Melatonin is known to bind calmodulin and directly affect the calcium signaling and modulate the activity of the enzymes such as adenylate cyclase and phosphodiesterase or structural proteins [28].

It is well established that Ca^2+^ channels play a vital role in the development and maintenance of central sensitization that is associated with inflammation and neuropathic pain, melatonin, therefore provides pain relief through modulation of the activity of Ca^2+^ channels [37,38].

### 4.4. Melatonin and Inflammatory Mediators

The inflammation is upregulated by various lipid mediators generated from arachidonic acid by the action of Phospholipase A_2_ (PLA_2_), and subsequent activity of 5-Lipoxygenase (5-LOX) and Cyclooxygenase-2 (COX-2). Melatonin can inhibit arachidonic acid formation, acting as a negative regulator of PLA_2_, inhibit the expression of 5-LOX and COX-2 leading to decreased synthesis of chemoattractant leukotrienes and prostaglandins, respectively [37,39]. Melatonin is also known to decrease the expression of iNOS (inducible nitric oxide synthase) and COX-2, which subsequently leads to a decline in NO and PGE_2_ expression, respectively [39]. These inflammatory mediators are involved in inflammatory pain perception and therefore, the inhibition of these inflammatory mediators contribute to the analgesic effects of melatonin (Figure 1).

## 5. Melatonin at the Crossroads of Different Chronic Pain Disorders, Sleep and Inflammation

Back pain is a commonly encountered musculoskeletal disorder that is often perceived to be related to disability. Most of the patients with acute back pain have good recovery, however, a small portion of the patients suffer from this pain over an extended period of time than acute back pain (more than 12 weeks) and are referred to have chronic back pain. Since, the incidence of acute back pain is higher, consequently the number of patients with chronic back pain is also getting higher [40]. It is estimated that 30–60% of the population in developed countries suffer from chronic back pain and it affects mostly the people of working age with a peak prevalence in the age group of 30–60 years. This represents a high cost to the society and contributes to disability [37,40]. It is of note that chronic back pain is responsible for the most years lived with disability when compared to any other medical illness [40].

Furthermore, melatonin provides pain relief in several other chronic pain conditions such as migraine, chronic cluster and tension type headaches, fibromyalgia, irritable bowel syndrome and chronic pelvic pain. Intriguingly, melatonin administration to chronic back pain patients has been shown to significantly relieve the pain both at rest and during motion [37,38]. In a clinical study, the addition of melatonin at a dosage of 3 mg/day, 30–40 min before sleeping at night, to the standard therapy consisting of administering glucosamine hydrochloride and chondroitin sulfate alone or in combination with diclofenac or only diclofenac was found to significantly reduce the pain intensity during movement and at rest as assessed by Visual Analog Scale (VAS) and related scoring batteries [37,38]. Melatonin addition to the standard regimens also provided additional benefits such as reduced pain-associated anxiety, mood stabilization and sleep quality improvement, which are often impaired in the chronic back pain patients [37]. These beneficial effects of melatonin might be due to the restoration of the circadian rhythm disturbed by the chronic pain, improvement in sleep and return of the body’s adaptive potential to normal besides the intrinsic analgesic properties of the melatonin administration. Intriguingly, in this study, the analgesic effects of melatonin were observed earlier than the sleep normalization effects, supporting the notion that the melatonin has its own intrinsic antinociceptive effects apart from similar effects brought about through complex interactions with other systems [37].

On one hand, the pain substantially affects the sleep and on the other hand, deprivation of restful sleep escalates the pain susceptibility [41]. So, both conditions can aggravate each other suggesting a bidirectional and reciprocal relationship, further adding to the suffering of the patient [42]. Chronic pain can result in sleep deprivation for extended periods of time and can affect the physical, emotional and behavioral wellbeing of the patients [41]. It is estimated that around 50–80% of the patients with chronic pain experience sleep disturbances [42]. Interestingly, a study involving chronic back pain patients showed that 53% of the patients met the criteria for insomnia compared to 3% of the healthy pain free controls [42]. Additionally, the severity of insomnia showed a positive association with pain intensity, sensory pain ratings, affective pain ratings, general anxiety, health anxiety and general depression. Clinical studies have shown that chronic pain patients reporting sleep disturbance have higher pain perception, more fatigue, poor mood, and increased levels of stress and disability. Alternatively, healthy subjects who were deprived of sleep showed an increased pain probably due to an increase in the release of pro-inflammatory cytokines and reduced pain tolerance [42].

Melatonin is shown to improve the quality of sleep in chronic inflammatory pain patients and other chronic pain disorders in addition to its analgesic effects [29,37]. Repeated administration of melatonin progressively improves the sleep and reduces anxiety, thereby leading to lower levels of pain [43]. Intriguingly, in an experimental chronic constriction injury rat model, deprivation of sleep was shown to further aggravate the neuropathic pain and was associated with increased activation of microglia and lower levels of serum melatonin [44]. However, when melatonin was administered, it attenuated the microglial activation, reduced the levels of pro-inflammatory cytokines ultimately leading to a reduction in neuropathic pain [44]. A study involving the administration of melatonin to chronic back pain patients also showed a significant improvement in the sleep quality in addition to a reduction in back pain [45].

A variety of inflammatory mediators are liberated upon tissue damage or inflammation including prostaglandins (PGE_2_), leukotrienes, bradykinin, substance P and inflammatory cytokines, which either directly activate nociceptors or release local allogenic agents capable of further sensitizing the nociceptors leading to enhanced pain transmission [30,46]. These inflammatory mediators setting the nociceptive system on fire are often produced by the immune cells. The coordination between nociceptive neuronal pathways and immune response represents a potential advantage to the host in protecting against danger (Figure 2). Interestingly, neuroimmune communication in pain is bidirectional, for instance immune cell secreted mediators sensitize peripheral and central pain pathways and in turn, nociceptive neurons actively release neuropeptides via their peripheral nerve terminals to shape the response of innate and adaptive immune cells [46].

Accumulating evidence suggests that melatonin can modulate the activation of immune system leading to a reduction in acute and chronic inflammation [3]. Melatonin demonstrates anti-inflammatory effects through the inhibition of the inflammatory cells activation by inhibiting myeloperoxidase activity [47]. Melatonin can produce anti-inflammatory effects through direct interaction with binding sites on macrophages and lymphocytes [39]. Melatonin inhibits the release of pro-inflammatory cytokines and downregulates the expression of NFκB in various experimental studies. Consequently, melatonin administration was found to inhibit the release of TNF-α from peritoneal macrophages obtained from mice (Figure 2) [39]. Melatonin has been shown to inhibit NLRP3 (nucleotide-binding oligomerization domain, leucine-rich-containing family, pyrin domain-containing-3) inflammasome activation under various conditions [47]. Interestingly, in an intervertebral disc degeneration (IVDD) rat model of low back pain, melatonin administration prevented IVDD progression and IVDD associated back pain, which was found to result from melatonin induced decrease in NLRP3, p20, and IL-1β levels due to inhibition of NFκB signaling and down modulation of mitochondrial ROS (reactive oxygen species) production [48]. In another neuropathic pain study, it was revealed that melatonin via MT2 receptors in the DRG (Dorsal root ganglion) could alleviate the allodynia and hyperalgesia in a sciatic nerve cuffing model of mice (Figure 2) [36]. Interestingly, melatonin inhibited the activation of peptogenic neurons and neuroinflammation in DRG through downregulation of inflammatory genes such as c-fos, calcitonin-gene related peptide (CGRP), TNF-α, IL-1β and nitric oxide synthase 1 (NOS1) [36].

In an Oxaliplatin induced neuropathic pain model, melatonin administration attenuated the astrocyte mediated spinal neuroinflammation and consequently, reduced pain hypersensitivity [49]. Another study found that melatonin administration significantly reduced the neuropathic pain in a chronic constrictive injury with sleep deprivation experimental model and exhibited significant inhibition of the microglial activation along with lower levels of inflammatory cytokines such as TNF-α, IL-1β and IL-6 (Figure 2) [44]. Interestingly, chronic constrictive injury of the median nerve in the cuneate nucleus led to the phosphorylation of p38 mitogen activated protein kinase (p38-MAPK) and microglial activation, which was associated with behavioral hypersensitivity [50]. Administration of melatonin reduced behavioral hypersensitivity, phosphorylation of p38-MAPK in microglia and consequently, reduced inflammatory cytokines secretion [50]. Intriguingly, a spinal nerve ligation (SNL) pain model was used to unveil the melatonin’s underlying epigenetic mechanisms of chronic pain alleviation by controlling the inflammatory genes expression [51]. SNL resulted in the reduced expression of the catalytic subunit of phosphatase 2A (PP2Ac) and increased histone deacetylase 4 (HDAC4) phosphorylation and cytoplasmic accumulation, which epigenetically relieved HDAC4-suppressed *hmgb1* gene transcription, resulting in enhanced high mobility group box-1 (HMGB-1) expression in the ipsilateral dorsal horn of the rats (Figure 2). Notably, melatonin acting via MT2 receptors upon intrathecal administration increased PP2Ac expression, HDAC4 dephosphorylation and nuclear accumulation, restored the HDCA4 mediated suppression of *hmgb1*, and thus, alleviated the SNL induced behavioral pain (Figure 2) [51].

All this evidence supports that melatonin can modulate the triad of chronic pain, sleep disturbance and inflammation. However, detailed clinical studies assessing the impact of Melatonin on levels of various inflammatory mediators in chronic back pain patients with disrupted sleep are lacking.

## 6. The Augmentative Potential of Melatonin for Neurostimulation Chronic Pain Therapies

Despite the origin and the marked differences in the pathophysiology, a broad range of chronic pain disorders have been linked to a disrupted sleep architecture. This may highlight the disturbance of the circadian pattern due to chronic pain disorders and may open the avenue for novel diagnostics and therapeutics taking advantage of the common and shared pathways of sleep and chronic pain [52,53,54,55,56,57,58,59,60,61,62]. In addition to a poor sleep, several chronic pain associated co-morbidities encompass obesity and related metabolic disorders such as hypertension, diabetes, atherosclerosis, cardiovascular pathologies and mood alterations (anxiety, depression, cognitive decline, pain catastrophizing). Notably, sleep deprivation was found to promote body weight gain in the long-term [63,64,65,66].

Based upon the pro-inflammatory profile of chronic pain conditions, several uncontrolled cohort and randomized-controlled neuromodulation trials determined concentration changes in a broad range of inflammatory mediators relevant for the genesis and maintenance of pain, sleep and metabolism in blood, saliva and cerebrospinal fluid [24,25,26,67,68,69]. In chronic back pain and neuropathic pain patients with a deteriorated sleep architecture and metabolic-associated co-morbidities such as obesity, cardiovascular ischemia, artherosclerosis, cerebral ischemia., increased blood levels of circulating IL-1β and leptin were detected indicating a pro-inflammatory state. These observations support the concept that chronic pain shall be defined as a single symptom among other characteristics (sleep, metabolism) of a more complex and dynamic multidimensional brain disorder affecting the brain and the peripheral nervous system by upregulating central and peripheral pro-inflammatory pathways. Interestingly, score-based metabolic phenotyping (body mass index; BMI) was found to be an outcome predictor for spinal cord stimulation in an observational pain study [70].

The use of melatonin in different chronic pain pathologies such as fibromyalgia, endometriosis, myofascial pain syndrome, primary headache disorders have yielded promising results supporting the analgesic capability of melatonin, although the number of studies and corresponding study design remains to be re-examined under controlled study conditions [71,72,73,74,75,76]. For instance, Danilov et al. found a significant decline in pain intensity and frequency along with an improved mood and sleep in a study cohort of chronic tension type headache patients treated with 3 mg melatonin at bedtime over 30 days [77].

In order to assess the complementary and synergistic effects of neurostimulation and melatonin, da Silva and co-workers combined melatonin with transcranial direct current stimulation of the motor cortex (tDCS) in healthy individuals using an acute pain paradigm and observed no superiority of combined melatonin and tDCS compared to sham treatment. However, melatonin alone significantly suppressed acute pain levels [78]. Whether, such adjunctive melatonin administration (combined with neurostimulation) will become useful to enhance chronic pain outcome remains currently unknown.

## 7. Conclusions

The adjunctive administration of melatonin to act synergistically with neurostimulation therapies for chronic pain management has yet to be evaluated in further clinical controlled pain trials applying either invasive and/or non-invasive approaches. Preliminary clinical findings support the idea to apply and assess the adjunctive impact of melatonin on chronic pain and associated co-morbidities, first to mention a pathological sleep pattern and pre-existing metabolic related diseases. From the preclinical perspective it appears to be reasonable to probe the value of complementary melatonin intake.

## Figures and Tables

**Figure 1 biomedicines-09-00984-f001:**
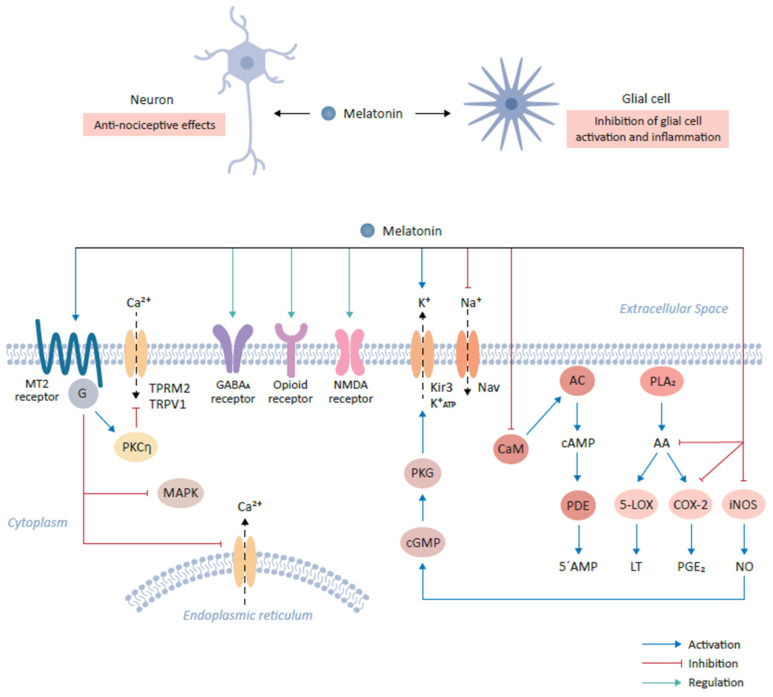
Role of melatonin and its cognate receptors in pain-relieving signal pathways. The binding of melatonin to MT2 receptors expressed on glutamatergic neurons mediates anti-nociceptive effects. This interaction results in the activation of the PKCη signaling pathway and in turn suppresses mitogen-activated protein kinase (MAPK) and calcium signaling. The interaction of melatonin with various neurotransmitter receptors, including gamma-aminobutyric acid (GABA)ergic (GABA_A_), produces anti-nociceptive effects. Furthermore, melatonin binds opioid and NMDA receptors to mediate downstream signaling, resulting in anti-nociceptive effects. Additionally, melatonin modulates ion flow by binding to Kir3 K^+^ channel which activates the outflow of K^+^ ions in the central nervous system. Conversely, melatonin inhibits the activities of sodium channels, including NAV1.8 and NAV1.9. Melatonin prevents the influx of Ca^2+^ by inhibiting calmodulin (CaM). CaM modulates intracellular 5’-AMP levels through the step-wise activation of adenylyl cyclase (AC) and cyclic nucleotide phosphodiesterase (PDE) by the intermediate generation of cAMP. Finally, melatonin modulates the inflammatory response and activation of glial cells by inhibiting the formation of arachidonic acid (AA) via the inhibition of phospholipase A_2_ (PLA_2_), which leads to the subsequent inhibition of 5-lipoxygenase (5-LOX) and cyclooxygenase-2 (COX-2) and the reduction in their targets, leukotriene (LT) and prostaglandin E2 (PGE_2_) respectively. Melatonin also decrease the expression of inducible nitric oxide synthase (iNOS), leading to the decline in nitric oxide (NO) expression.

**Figure 2 biomedicines-09-00984-f002:**
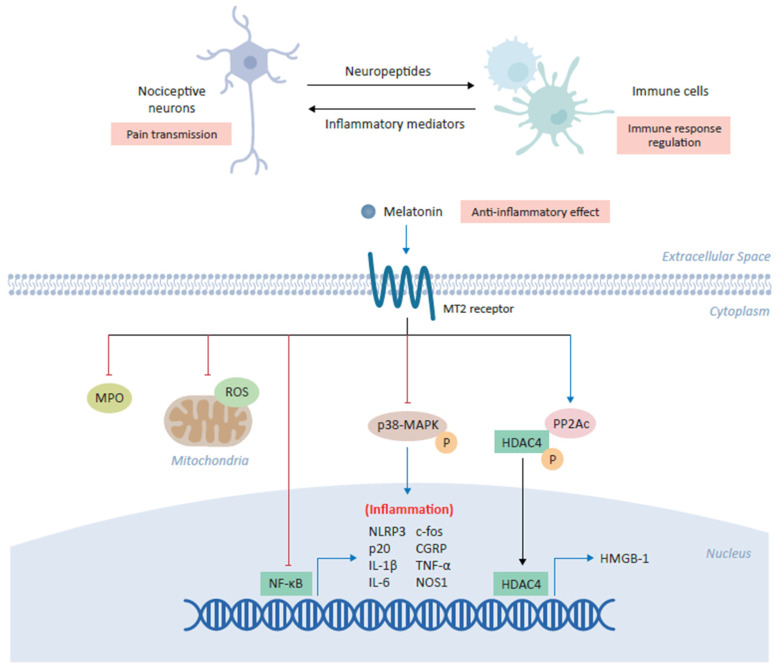
Anti-inflammatory role of melatonin in alleviating chronic pain. Melatonin plays a central role in the coordination between nociceptive neurons and immune cells in which its binding the cross-talk between inflammation and pain transmission. The binding of melatonin to MT2 receptors expressed on immune cells inhibits myeloperoxidase (MPO) activity and NF-ĸB, as well as downregulating the production of mitochondrial reactive oxygen species (ROS). Melatonin has also been shown to inhibit the NLRP3 (NOD-, LRR- and pyrin domain-containing protein 3) inflammasome and modulate the downstream expression of several inflammatory genes, including c-fos, calcitonin-gene-related peptide (CGRP), IL-1β, IL-6, TNF-α and NOS1. In various cell types, melatonin inhibits the phosphorylation of p38 mitogen-activated protein kinase (MAPK). Finally, melatonin increases expression of the catalytic subunit of phosphatase 2A (PP2Ac) which in turn mediates the dephosphorylation of HDAC4, driving its nuclear accumulation to repress the expression of high mobility group box-1 (HMGB-1) and ultimately alleviating chronic pain.

## Data Availability

Data available on request due to restrictions eg privacy or ethical The data presented in this study are available on request from the corresponding author.

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
