# Peer review of "Melatonin Moderates the Triangle of Chronic Pain, Sleep Architecture and Immunometabolic Traffic"

_biomedicines, 2021, doi:10.3390/biomedicines9080984_

Round 1

Reviewer 1 Report

The present review by Chaudhry et al. could be really considered as a review of reviews, as many of the references that are commented in this review are previous reviews of other authors.

It is curious that authors indicate “due to the limited data available (??)..... the findings were analyzed in a narrative review instead of a meta-analytic approach”. This seems to be true, as data are reviewed like in a random order, particularly when authors try to correlate pain, sleep, immune system and metabolism (obesity ??)... For instance, in paragraph 5, they initially describe effects of melatonin on chronic back pain, then effects in other types of neuropathic pain, then sleep deprivation, then inflammatory mediators release, modulation of immune system.... them go back to neuropathic pain....

Also in paragraph 6, where authors try to indicate the benefits of melatonin on neurostimulation related to chronic pain therapy, in which authors correlate chronic pain, with sleep disorders and weight gain (??), leading to the conclusion about the non superiority of combined melatonin and neurostimulation and melatonin, or that that “Whether therapeutic adjunctive approach will become useful to treat chronic pain remains currently largely unknown”.

Other minor (?) points

-Page 3: Authors state that “Melatonin is also known to upregulate the expression...... and glutathione peroxidase (Figure 1)”. No indication to these enzymes and or their overexpression is shown in Figure 1.

-Page 5: Authors state  that “Flumazenil, which is a benzodiazepine antagonists and consequently GABAA receptor antagonists”. From a pharmacological point of view, a benzodiazepine receptor antagonist (i.e. Flumazenil) can not be considered as a GABAA receptor antagonists, as it does not directly blocks the effects of GABA on this receptor subtype. Flumazenil antagonize the allosteric effects of benzodiazepines on their own receptor (a different binding site in the GABA receptor).

-There are multiple small typographical mistakes along the text (i.e.: capbility, trafficing, proporties, neuromdulation, antinocioceptive, nocioceptors, nocioceptive, communcation, re-exmained.....) 

Author Response

We want to express out thankfulness for the efforts and time both reviewer ́s underwent.

REVIEWER 1

The present review by Chaudhry et al. could be really considered as a review of reviews, as many of the references that are commented in this review are previous reviews of other authors.

It is curious that authors indicate “due to the limited data available (??)..... the findings were analyzed in a narrative review instead of a meta-analytic approach”. This seems to be true, as data are reviewed like in a random order, particularly when authors try to correlate pain, sleep, immune system and metabolism (obesity ??)...

Response: We have modified the sentences in order to clarify the intended meaning:

Chronic pain studies assessing and combining neurostimulation with melatonin are lacking, while there is exist a large number of publication addressed to melatonin neurobiology indicating melatonin to play a crucial role for the pathophysiology of chronic pain, sleep and metabolism. All these characteristics are disrupted frequently in chronic pain patients. In particular, the impact and adjunct value of melatonin for neurostimulation therapies intended to treat chronic pain disorders remains largely unknown. Due to the limited data available pertaining to the particular pain types along with outcome parameters, the findings were presented in the form of a narrative review.

For instance, in paragraph 5, they initially describe effects of melatonin on chronic back pain, then effects in other types of neuropathic pain, then sleep deprivation, then inflammatory mediators release, modulation of immune system.... them go back to neuropathic pain....

Also in paragraph 6, where authors try to indicate the benefits of melatonin on neurostimulation related to chronic pain therapy, in which authors correlate chronic pain, with sleep disorders and weight gain (??), leading to the conclusion about the non superiority of combined melatonin and neurostimulation and melatonin, or that that “Whether therapeutic adjunctive approach will become useful to treat chronic pain remains currently largely unknown”.

Response: We intended to show the relationship melatonin has for different pain disorders such as back pain, neuropathic pain or primary headache disorders. From the clinical phenotypes, chronic pain patients suffers from sleep disturbance and metabolic-associated disorders. This is indicative that for shared relevant pathways by neuroinflammation means.

We have modified the paragraph 5 and paragraph 6. Corresponding changes were made in the subheading and different sections of paragraph 5 and 6. Changes were marked blue.

Other minor (?) points

-Page 3: Authors state that “Melatonin is also known to upregulate the expression...... and glutathione peroxidase (Figure 1)”. No indication to these enzymes and or their overexpression is shown in Figure 1.

Response: In line with the reviewer, we revised corresponding paragraphs on page 3 and on page 4. Changes were marked blue.

-Page 5: Authors state that “Flumazenil, which is a benzodiazepine antagonists and consequently GABAA receptor antagonists”. From a pharmacological point of view, a benzodiazepine receptor antagonist (i.e. Flumazenil) can not be considered as a GABAA receptor antagonists, as it does not directly blocks the effects of GABA on this receptor subtype. Flumazenil antagonize the allosteric effects of benzodiazepines on their own receptor (a different binding site in the GABA receptor).

Response: We have corrected the sentences

Melatonin can modulate the GABAergic transmission as was evidenced from the observation that Flumazenil, which is a benzodiazepine antagonist and inhibits indirectly the GABAergic neurotransmission via antagonizing the allosteric effects of benzodiazepines on their own receptors subtype and therefore, reversing the anti-nociceptive effects of Melatonin [32,37,38,39].

-There are multiple small typographical mistakes along the text (i.e.: capbility, trafficing, proporties, neuromdulation, antinocioceptive, nocioceptors, nocioceptive, communcation, re- exmained.....)

Response: We carefully re-examined the entire manuscript for proof-reading and in addition, a Native English speaker revised the manuscript in a second round.

Reviewer 2 Report

In the Review entitled “Melatonin moderates the triangle of chronic pain, sleep disruption and immunometabolic traffic and may augment neurostimulation for chronic pain” the authors showed the impact of melatonin on the crossroad of sleep, chronic pain and immunometabolism. The authors do a good job of bibliographic research on melatonin and chronic pain and the correlation with sleep disorders and immunometabolism is interesting. Unfortunately, the paper requires an important overall revision and above all a more correct approach to the construction of the sentences that thus formulated do not make reading easier.

Title: What this part of the title means: “may augment neu- rostimulation for chronic pain” The first part of the title was sufficient

Introduction page 2: The whole paragraph that begins with the sentence: “On the other hand non-invasive and invasive neurostimulation therapies of the central and peripheral nervous system have become an established therapy option in other- wise drug-resistant chronic pain disorders of different origin including but not limited to spinal cord stimulation, dorsal root ganglion stimulation and/or vagus nerv stimulation” is unclear

“3.1. Melatonin and its receptors”.  It is a sub-paragraph of the paragraph “3. Synthesis and metabolism of Melatonin”?

 “3.2. MT1 and MT2 receptors” It is another sub-paragraph different from 3.1?

  1. Melatonin and Chronic Pain : “Alternatively, analgesia observed after melatonin administration in both nociceptive and neuropathic pain models also implicates an important role of melatonin in regulating pain” Because the sentence begins with alternatively?

4.1 ”The pain relieving effects of melatonin either stem from its direct anti-nociceptive effects owing to its anti-oxidant, anti-inflammatory properties and direct interaction with its cognate receptors; or through indirect complex interactions with other systems that can modulate the pain perception” This is one of many examples of a convoluted sentence

4.3 “Melatonin also inhibits the voltage-dependent neuronal inflow of Ca2+ currents in the cultured dorsal root ganglion cells of the spinal cord, which stems from the Melatonin’s ability to block voltage depend- ent Ca2+ channels leading to a decline in the free intracellular Ca2+ concentration” Another example of an unclear sentence.

5. In this paragraph the authors report numerous data on melatonin and chronic pain, melatonin and inflammatory mediators… with numerous repeated phrases. This data was to be part of the respective paragraphs. Here the authors should have put a focus on “Melatonin at the crossroads of chronic pain, sleep and immunometabolism” as well as the title of the paragraph promised

Author Response

REVIEWER 2

We want to thank the reviewer for his valuable comments and concerns, which we have addressed point-by-point

In the Review entitled “Melatonin moderates the triangle of chronic pain, sleep disruption and immunometabolic traffic and may augment neurostimulation for chronic pain” the authors showed the impact of melatonin on the crossroad of sleep, chronic pain and immunometabolism. The authors do a good job of bibliographic research on melatonin and chronic pain and the correlation with sleep disorders and immunometabolism is interesting. Unfortunately, the paper requires an important overall revision and above all a more correct approach to the construction of the sentences that thus formulated do not make reading easier.

Title: What this part of the title means: “may augment neu- rostimulation for chronic pain” The first part of the title was sufficient

Response: We revised the title and deleted the second part.

Melatonin moderates the triangle of chronic pain, sleep disruption and immunometabolic traffic 

Introduction page 2: The whole paragraph that begins with the sentence: “On the other hand non-invasive and invasive neurostimulation therapies of the central and peripheral nervous system have become an established therapy option in other- wise drug-resistant chronic pain disorders of different origin including but not limited to spinal cord stimulation, dorsal root ganglion stimulation and/or vagus nerv stimulation” is unclear

Response: We revised the entire paragraph according to the reviewer comment. Corresponding changes were marked blue.

“3.1. Melatonin and its receptors”.  It is a sub-paragraph of the paragraph “3. Synthesis and metabolism of Melatonin”?

Response: In line with the suggestion of the reviewer, we have modified numbering of the headings.

 “3.2. MT1 and MT2 receptors” It is another sub-paragraph different from 3.1?

Response: In line with the suggestion of the reviewer, we have modified numbering of the headings.

  1. Melatonin and Chronic Pain : “Alternatively, analgesia observed after melatonin administration in both nociceptive and neuropathic pain models also implicates an important role of melatonin in regulating pain” Because the sentence begins with alternatively?

Response: In line with the suggestion of the reviewer, we have modified the sentence as follows:

‘Further adding to the evidence, analgesia observed after melatonin administration in both nociceptive and neuropathic pain models also points towards an important role of melatonin in regulating pain’.

4.1 ”The pain relieving effects of melatonin either stem from its direct anti-nociceptive effects owing to its anti-oxidant, anti-inflammatory properties and direct interaction with its cognate receptors; or through indirect complex interactions with other systems that can modulate the pain perception” This is one of many examples of a convoluted sentence

Response: In line with the suggestion of the reviewer, we have modified the sentence as follows:

‘The pain relieving effects of melatonin either stem from its direct anti-nociceptive effects due to its anti-oxidant, anti-inflammatory properties and direct inter-action with its cognate receptors; or through indirect complex interactions with other systems that can modulate the pain perception pathways’.

4.3 “Melatonin also inhibits the voltage-dependent neuronal inflow of Ca2+ currents in the cultured dorsal root ganglion cells of the spinal cord, which stems from the Melatonin’s ability to block voltage depend- ent Ca2+ channels leading to a decline in the free intracellular Ca2+ concentration” Another example of an unclear sentence.

Response: In line with the suggestion of the reviewer, we have modified the sentence as follows:

‘In the cultured dorsal root ganglion cells of the spinal cord, Melatonin has also been shown to decrease the neuronal free intracellular Ca2+ levels via inhibition of the volt-age sensitive Ca2+ channels’.

  1. In this paragraph the authors report numerous data on melatonin and chronic pain, melatonin and inflammatory mediators… with numerous repeated phrases. This data was to be part of the respective paragraphs. Here the authors should have put a focus on “Melatonin at the crossroads of chronic pain, sleep and immunometabolism” as well as the title of the paragraph promised

Response: In line with the reviewer’s suggestion, we have removed the repeating phrases. We have modified the title of the heading and this portion was specifically focused to discuss the role of inflammation in detail during different chronic pain conditions and the impact of melatonin to relieve the pain, improve sleep and decrease inflammation.

Round 2

Reviewer 2 Report

English language and style are fine/minor spell check required